# The growth of the central region by acquisition of counterrotating gas in star-forming galaxies

Yan-Mei Chen[1,2,3], Yong Shi[1,2,3], Christy A. Tremonti[4], Matt Bershady[4], Michael Merrifield[5], Eric Emsellem[6,7], Yi-Fei Jin[1,2,3], Song Huang[8], Hai Fu[9], David A. Wake[10], Kevin Bundy[8], David Stark[8], Lihwai Lin[11], Maria Argudo-Fernandez[12,13], Thaisa Storchi Bergmann[14,15], Dmitry Bizyaev[16,17], Joel Brownstein[18], Martin Bureau[19], John Chisholm[4], Niv Drory[20], Qi Guo[21], Lei Hao[12], Jian Hu[22,23], Cheng Li[22,23], Ran Li[21], Alexandre Roman Lopes[24], Kai-Ke Pan[16], Rogemar A. Riffel[15,25], Daniel Thomas[26], Lan Wang[21], Kyle Westfall[26] & Ren-Bin Yan[27]

Galaxies grow through both internal and external processes. In about 10% of nearby red galaxies with little star formation, gas and stars are counter-rotating, demonstrating the importance of external gas acquisition in these galaxies. However, systematic studies of such phenomena in blue, star-forming galaxies are rare, leaving uncertain the role of external gas acquisition in driving evolution of blue galaxies. Here, based on new measurements with integral field spectroscopy of a large representative galaxy sample, we find an appreciable fraction of counter-rotators among blue galaxies (9 out of 489 galaxies). The central regions of blue counter-rotators show younger stellar populations and more intense, ongoing star formation than their outer parts, indicating ongoing growth of the central regions. The result offers observational evidence that the acquisition of external gas in blue galaxies is possible; the interaction with pre-existing gas funnels the gas into nuclear regions ($<1$ kpc) to form new stars.

[1] School of Astronomy and Space Science, Nanjing University, Nanjing 210093, China. [2] Key Laboratory of Modern Astronomy and Astrophysics (Nanjing University), Ministry of Education, Nanjing 210093, China. [3] Collaborative Innovation Center of Modern Astronomy and Space Exploration, Nanjing 210093, China. [4] Department of Astronomy, University of Wisconsin-Madison, 1150 University Ave, Madison, Wisconsin 53706, USA. [5] School of Physics and Astronomy, University of Nottingham, University Park, Nottingham NG7 2RD, UK. [6] European Southern Observatory, Karl-Schwarzschild-Strasse 2, D-85748 Garching, Germany. [7] Université Lyon 1, Observatoire de Lyon, Centre de Recherche Astrophysique de Lyon and Ecole Normale Supérieure de Lyon, 9 avenue Charles André, F-69230 Saint-Genis Laval, France. [8] Kavli Institute for the Physics and Mathematics of the Universe (Kavli IPMU, WPI), Todai Institutes for Advanced Study, the University of Tokyo, Kashiwa 277-8583, Japan. [9] Department of Physics and Astronomy, University of Iowa, Iowa City, Iowa 52242, USA. [10] Department of Physical Sciences, The Open University, Milton Keynes MK7 6AA, UK. [11] Institute of Astronomy and Astrophysics, Academia Sinica, Taipei 106, Taiwan. [12] Shanghai Astronomical Observatory, Nandan Road 80, Shanghai 200030, China. [13] Universidad de Antofagasta, Unidad de Astronoma, Facultad Cs. Bsicas, Av. U. de Antofagasta, 02800 Antofagasta, Chile. [14] Departamento de Astronomia, Instituto de Física, Universidade Federal do Rio Grande do Sul, CP 15051, 91501-970, Porto Alegre, RS, Brazil. [15] Laboratório Interinstitucional de e-Astronomia—LlneA, Rua Gal. José Cristino 77, Rio de Janeiro, RJ 20921-400, Brazil. [16] Apache Point Observatory and New Mexico State University, P.O. Box 59, Sunspot, New Mexico 88349-0059, USA. [17] Sternberg Astronomical Institute, Moscow State University, Moscow 119899, Russia. [18] Department of Physics and Astronomy, University of Utah, Salt Lake City, Utah 84112, USA. [19] Sub-Department of Astrophysics, University of Oxford, Denys Wilkinson Building, Keble Road, Oxford OX1 3RH, UK. [20] Department of Astronomy and Astrophysics, University of California, Santa Cruz, California 95064, USA. [21] National Astronomical Observatories, Chinese Academy of Sciences, 20A Datun Road, Chaoyang, Beijing 10012, China. [22] Department of Physics, Tsinghua University, Beijing 100084, China. [23] Center for Astrophysics, Tsinghua University, Beijing 100084, China. [24] Departamento de Fsica, Facultad de Ciencias, Universidad de La Serena, Cisternas 1200, La Serena, Chile. [25] Departamento de Física, Centro de Ciências Naturais e Exatas, Universidade Federal de Santa Maria, 97105-900 Santa Maria, RS, Brazil. [26] Institute for Cosmology and Gravitation, University of Portsmouth, Dennis Sciama Building, Burnaby Road, Portsmouth PO1 3FX, UK. [27] Department of Physics and Astronomy, University of Kentucky, 505 Rose Street, Lexington, Kentucky 40506-0055, USA. Correspondence and requests for materials should be addressed to Y.-M.C. (email: chenym@nju.edu.cn).

In the framework of hierarchical structure formation, a galaxy grows from primordial density fluctuations and its subsequent evolution is shaped by a series of external and internal processes. Galaxies with gas and stars counter-rotating are the key demonstrations for the regulation by external processes[1,2]. External processes, for example major mergers, minor mergers or gas accretion, could bring gas which is counter-rotating with pre-existing stars into the galaxies. On the other hand, the gas produced by internal processes such as stellar evolution would conserve the angular momentum of stars and be co-rotating with pre-existing stars.

Phenomenon of gas and star counter-rotating is now known to be ubiquitous in elliptical and lenticular galaxies. Still, the incidence of gas-star counter-rotators in blue star forming galaxies is largely unknown. Since the early discoveries of individual cases[3], systematic studies with long-slit spectroscopy have reported a fraction as high as 25% (refs 4–6) in early type galaxies, which decreased to a value of 10–15% with integral-field spectroscopy[7–9]. While a few individual cases of blue counter-rotators are found[10–13], existing statistical studies of blue galaxies failed to identify any blue counter-rotators due to limited sample size[6,14] and instrumentation (for example, the limited ability of long-slit spectroscopy to effectively identify the pattern of the star-gas counter-rotating out of complicated kinematics, particularly in barred spirals[15]).

To place much stronger constraints on the incidence of blue counter-rotators, and to understand the influence of gas accretion on the evolution of blue star forming galaxies, in this work we study a sample of galaxies observed with fibre-optic integral-field units (IFU) in the first year of the survey: Mapping Nearby Galaxies at Apache Point Observatory (MaNGA)[16], finding ~2% blue star-forming galaxies have counter-rotating gas. The central regions of blue counter-rotators show younger stellar populations and more intense, ongoing star formation than their outer-skirts, indicating that these galaxies accrete abundant external gas, the interaction with pre-existing gas triggers the gas into central regions and form new stars.

## Results

**Sample selection**. We analyse gas and stellar kinematic maps of a representative sample of 1,351 nearby galaxies with stellar masses above $10^9$ solar mass from MaNGA. Figure 1 shows an example of a counter-rotating blue star forming galaxy. The Sloan Digital Sky Survey (SDSS) false-colour image is at left, while the kinematics based on spectroscopic IFU data for stars and gas are mapped in the second and third columns (velocities and velocity dispersions, respectively). To quantify the kinematic misalignment between stars and gas, we measured the difference in the kinematic position angle (PA) between ionized gas and stars as $\Delta PA = |PA_* - PA_{gas}|$, where $PA_*$ is the PA of stars and $PA_{gas}$ is the PA of ionized gas. The kinematic PA is measured based on established methods[17], defined as the counter-clockwise angle between north and a line that bisects the velocity field of gas or stars, measured on the receding side. The solid lines in Fig. 1 show the best fit position angle and the two dashed lines show the $\pm 1\sigma$ error. The last two columns show the rotation velocity and velocity dispersion along the major axis.

We matched the MaNGA sample with the literature catalogue[18] to obtain the global star formation rate (SFR) and stellar mass ($M_*$) for 1,220 out of 1,351 galaxies. With these two quantities we classify the sample into blue star-forming galaxies, red quiescent galaxies with little star formation and green-valley galaxies between these two extremes (Fig. 2a), as summarized in Table 1. For simplicity, we refer to these three classes as blue, red and green galaxies henceforth. Figure 2b shows the distributions of $\Delta PA$ for these different types of galaxies with nebular emission (required to measure the gas kinematics). Both green (green histogram) and red (red histogram) galaxies have a distribution of the $\Delta PA$, with the three local peaks at $\Delta PA = 0°$, $90°$ and $150°$, while blue galaxies (blue histogram) present a bimodal distribution (the lack of a third peak at $90°$ being consistent with small number statistics). The grey histogram is for the whole population—the combination of blue, red and green. In total there are 43 counter-rotators, that is, galaxies with $\Delta PA > 150°$. Considering the completeness correction of the MaNGA sample, the fraction of the counter rotators in blue galaxies is 2% (9 out of 489), while the fractions in red and green galaxies are 10% (16 out of 164) and 6% (18 out of 280), respectively. Our fraction of counter-rotators in the red galaxies is consistent with previous studies[4,5,7–9,19]. Thanks to the unbiased MaNGA galaxy sample with respect to morphology, inclination, colour and so on, we can study the incidence as well as the properties of blue counter-rotators for the first time. The above fractions could be lower limits, since for face-on galaxies, it is not possible to measure rotation.

**Properties of blue star-forming counter-rotators**. Among nine blue counter-rotators, six of them have strong positive gradients in the 4,000 Å break (D4000), as shown in Fig. 3, while the remaining show small D4000 across the whole galaxy body,

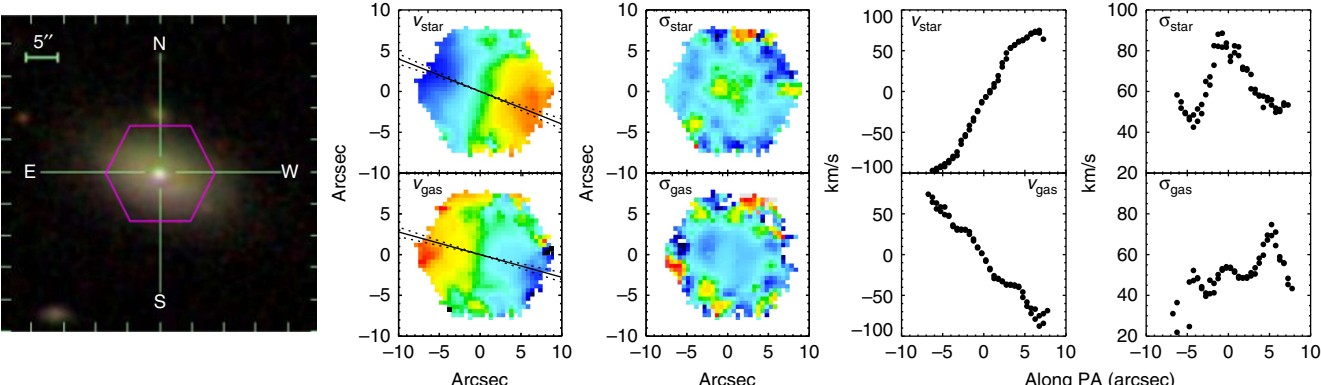

**Figure 1 | An example of a blue star-forming counter-rotating galaxy.** The left panel shows the SDSS g, r, i—band image, the projected velocity fields of stars (top) and gas (bottom) are shown in the second column, while the third column shows the velocity dispersion maps of stars and gas. The projected velocity and velocity dispersion along major axis (black solid line in the second column) are shown in the last two columns. Dashed black lines represent $\pm 1\sigma$ uncertainties in the major-axis position angle.

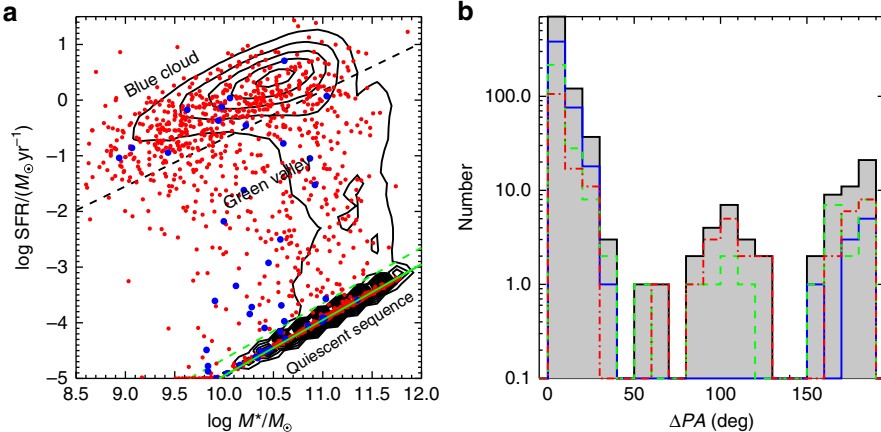

**Figure 2 | SFRs versus stellar masses and ΔPA distribution. (a)** SFRs versus stellar mass. Contours show the SDSS DR7 sample, while the red dots are MaNGA galaxies. The blue dots are the counter-rotators with $\Delta PA > 150°$. The two dashed lines separate the galaxies into blue star-formers, green valley and red quiescent galaxies. The black dashed line is adopted from Fig. 11 of ref. 18 as an approximation of the boundary (at the $1\sigma$ level in scatter) of the star-forming main sequence. The green solid line with log sSFR ($\equiv$SFR/M$\star$) $\sim -15$ remarks red galaxies, in which the SFR can be neglected. The region between the black and green dashed lines is referred as the green valley. Although galaxies in the green valley have low SFR, they are clearly distinguished from red galaxies. We do not use the colour-magnitude diagram to separate blue from green and red galaxies since the colours are strongly effected by dust extinction. **(b)** ΔPA distribution for MaNGA galaxies with nebular emission. The grey histogram is for the whole sample, red for the red quiescent galaxies.

**Table 1 | Classification of the MaNGA sample.**

| Type | Number (number with EML) | Misalignment | Counter-rotators |
|---|---|---|---|
| | | ($\Delta PA > 30°$) | ($\Delta PA > 150°$) |
| Blue | 489 (489) | 10 | 9 |
| Green | 377 (280) | 26 | 18 |
| Red | 354 (164) | 30 | 16 |
| Total | 1220 (933) | 66 | 43 |

This table gives the number of galaxies in each catagory. blue: blue star forming galaxies; green: green valley; red: red quiescent galaxies. Misalignment and counter-rotators are classified by ΔPA given in the table. EML means galaxies with emission lines; the number of galaxies with line emission is in parenthesis.

indicating young stellar populations existing in the central regions. The map of the Hα flux further shows ongoing star formation in the central region. We checked the emission line ratio diagnostic[20] to assure that the Hα radiation is dominated by star formation instead of active galactic nuclei (AGN; Fig. 5). In contrast to the blue counter-rotators, all the green and red counter-rotators have negative D4000 gradients with older stellar populations in the central regions. Although the Hα flux also peaks at the center for the green and red counter rotators, it is primarily contributed by the AGN based on the emission-line diagnostic[20].

To further quantify the importance of the ongoing star formation in growing the central region, we introduce the star formation activity parameter[21] as $\alpha_{SF} = 1/(sSFR \times (t_H(z) - 1\,\mathrm{Gyr}))$, where $t_H(z)$ is the Hubble time at the redshift of the galaxy, and 1 Gyr is subtracted to account for the fact that star formation mainly occurred after reionization. If a galaxy's current SFR is equal to its past average ($M_*/((t_H(z) - 1\,\mathrm{Gyr}))$) then $\alpha_{SF} = 1$; values less than one indicate that the current SFR is higher than the past average. As shown in Fig. 4, all nine galaxies present a steep rising $\alpha_{SF}$ with increasing distances from the galaxy center. The grey shaded regions show the $\pm 1\sigma$ range of $\alpha_{SF}$ for the central 1 kpc of local star forming galaxies with $\Delta PA < 30°$. Grey lines mark the median value of $\sim 0.75$. Focusing on the central 1 kpc, we find six of the blue counter rotators have $\alpha_{SF}$ about one order of magnitude

smaller than the average value (the grey line), indicating fast growth of the central components of these galaxies.

Both the D4000 and star formation activity parameter $\alpha_{SF}$ suggest significant ongoing growth of the central region ($< 1$ kpc) of these blue counter-rotators by star formation. For nine blue counter-rotators, we fit the r-band surface brightness profiles (Figs 6 and 7) and found that five of them already have photometric bulge-like components (above an exponential disk-like component). In addition, the SDSS images show no signs of strong galaxy interactions or major merging, indicating accretion of gas from intergalactic medium or dwarfs (minor mergers) as the origin of the counter-rotating gas. This is also consistent with their environments, as both the neighbour number ($N$) and the tidal strength parameter[22] ($Q_{lss}$) indicate that the blue counter rotators tend to be located in more isolated environments. By matching our galaxies with the MPA-JHU catalogue (http://wwwmpa.mpa-garching.mpg.de/SDSS/DR7/oh.html), we obtained the metallicity for eight blue counter-rotators. Four of them follow the stellar mass versus metallicity relation of the general population[23], while another four lie 0.2–0.3 dex above the stellar-mass versus metallicity relation.

## Discussion

We suggest the following scenario to explain the above observational facts: (i) The progenitor accretes counter-rotating gas from a gas-rich dwarf or cosmic web. (ii) Redistribution of angular momentum occurs from gas–gas collisions between the pre-existing and the accreted gas largely accelerates gas inflow, leading to a fast centrally-concentrated star formation. (iii) Higher metallicity is a puzzle, one possibility is due to the enrichment from star formation. In a closed-box model[24], the metallicity will mainly depend on the gas mass fraction $f_{gas}$ ($\equiv M_{gas}/(M_{gas} + M_{stars})$), so the abundances get elevated instantaneously as a large fraction of the available gas turns into stars. The low D4000 at the center is a hint that such stars exist. However, we keep in mind that the 'external' gas likely had low metallicity and the closed-box model is a strong assumption, future simulations are necessary in helping us to understand the gas enrich process.

Though the amount of pre-existing and accreted gas in the nine galaxies is uncertain, collision between pre-existing and

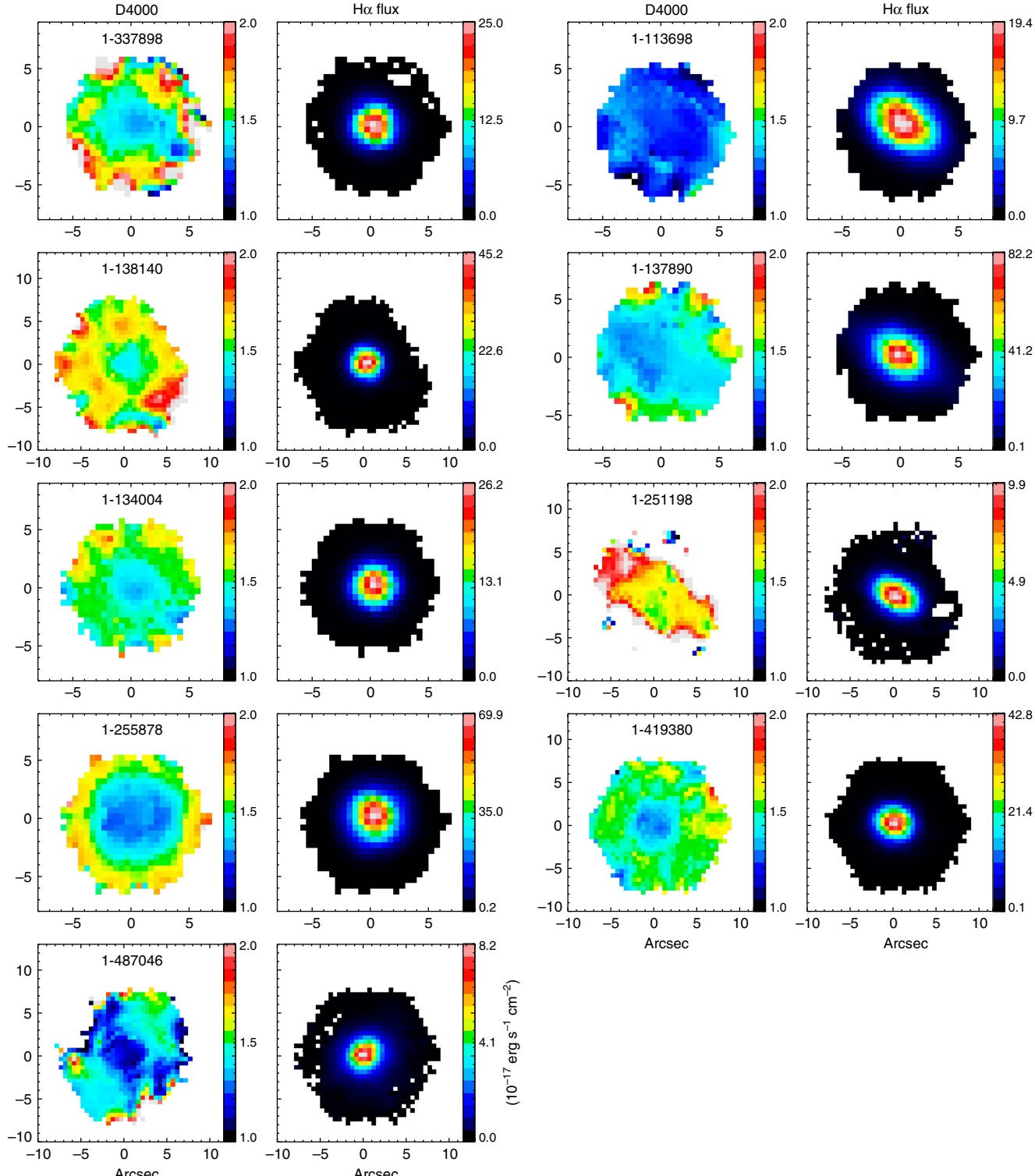

**Figure 3 | The D4000 and Hα flux maps for nine star-forming counter rotators.** The MaNGA-ID for each galaxy is shown in the D4000 map. The Hα flux is in the unit of $10^{-17}\,erg\,s^{-1}\,cm^{-2}$.

accreted gas is unavoidable, leading to redistribution of angular momentum and dissipation of kinetic energy. The impact on both the morphology and dynamics of the inner parts of the galaxy may thus be associated with the observed slight increase of the gas velocity dispersion. We find the typical gas velocity dispersion ($40$–$60\,km\,s^{-1}$) in the disk region of these nine galaxies is about $20\,km\,s^{-1}$ larger than a control sample of star-forming galaxies

with aligned gas and stellar kinematics ($\Delta PA < 30°$), closely matched in SFR, $M_*$ and redshift. The typical errors of gas velocity dispersion is about $10\,km\,s^{-1}$.

In summary, redistribution of angular momentum through the collisions between accreted and pre-existing gas is thus an efficient way for gas to migrate to the centre, indicating that accretion of counter-rotating gas into disk galaxies is an effective

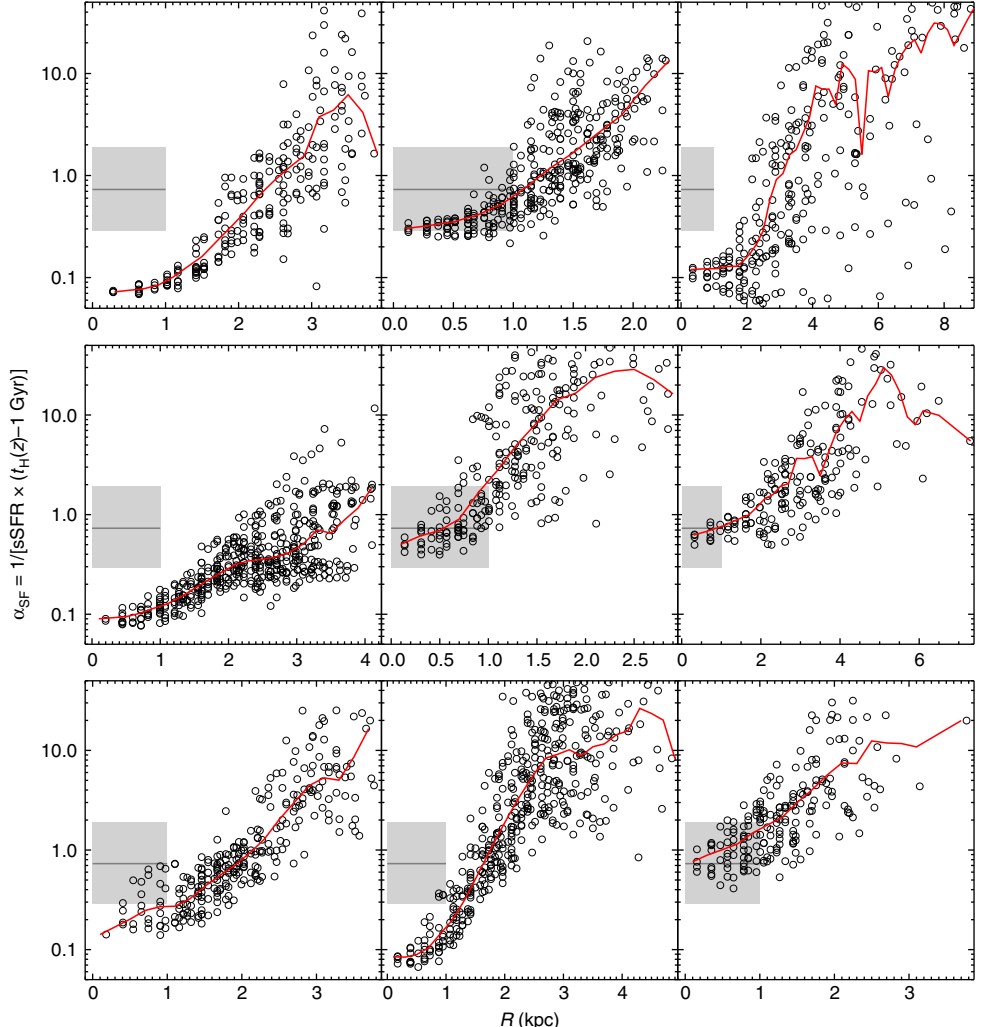

**Figure 4 | The star formation activity parameter $\alpha_{SF}$ versus radius for the nine blue star forming counter rotators.** The circles are our data points while the red lines show the median. The grey shaded regions show the $\pm 1\sigma$ range of $\alpha_{SF}$ for the central 1kpc of local star forming galaxies with $\Delta PA < 30°$. Grey lines mark the median value of $\sim 0.75$.

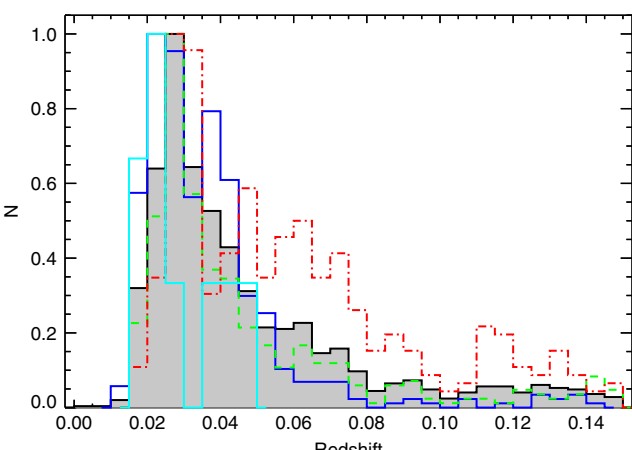

**Figure 5 | The redshift distributions of the samples.** The grey histogram is for the whole MaNGA sample; the blue, red and green histograms show the redshift distributions for the blue, red and green sub-samples, respectively; the nine blue counter rotators are shown in cyan histogram.

way to grow the central region. This mechanism may be more effective in growing the central component of galaxies at $z \sim 1$–$2$ where external gas acquisition is more frequent[25,26].

## Methods

**Observations and data reduction.** The data used in this work comes from the ongoing MaNGA survey[16,27–29] using the SDSS 2.5-in telescope[30] and Baryon oscillation spectroscopic survey spectrographs[31]. As one of three programs comprising the SDSS-IV, MaNGA is obtaining spatially resolved spectroscopy for about 10,000 nearby galaxies with log $M_*/M_\odot \geq 9$, and a median redshift of $z \approx 0.04$. The $r$-band signal-to-noise ratio (S/N) in the outskirts of MaNGA galaxies is 4–8 Å$^{-1}$, and the wavelength coverage is $3,600 - 10,300$ Å. MaNGA's effective spatial and spectral resolution is $2''.4$ (Full Width at Half Maximum, FWHM) and $\sigma \sim 60 \, \text{km s}^{-1}$, respectively. The MaNGA sample and data products used here were drawn from the internal MaNGA Product Launch-4 (MPL-4), which includes $\approx 1,400$ galaxies observed through July 2015 (the first year of the survey).

The MaNGA data analysis pipeline, which uses pPXF[32] and the MIUSCAT stellar library[33], fits the stellar continuum in each spaxel and produces estimates of the stellar kinematics. Ionized gas kinematics, $v_{gas}$ and $\sigma_{gas}$, as well as the flux were estimated by fitting a single Gaussian to the emission lines after stellar continuum subtraction. The observables used in this work, that is, $v_{gas}$ and $\sigma_{gas}$, D4000, emission line flux, are from data analysis pipeline.

**Redshift distributions of the samples.** In Fig. 5, we show the redshift distributions of the whole MaNGA sample (black histogram), the blue (blue histogram),

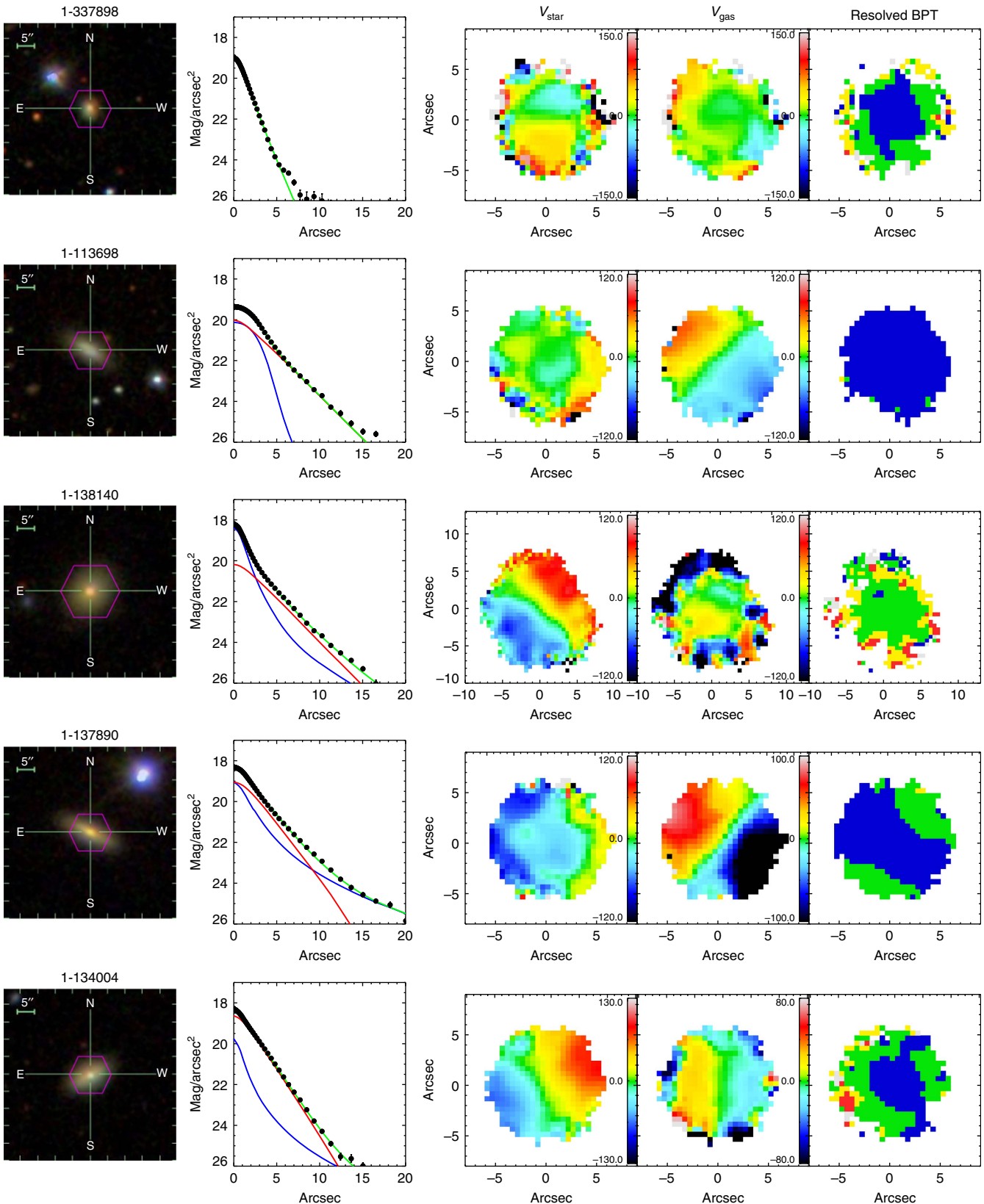

**Figure 6 | Properties of the blue counter rotators.** Left: the SDSS false-colour image; second column: the surface brightness profile, black is the data, green is the best fit model. Except for the first object, all the others are fitted by two components (red + blue); the third and fourth columns show the velocity fields of stars and gas, respectively. The velocities are in the unit of km s$^{-1}$. The spatial resolved BPT diagram[20] is shown in the last column, blue represents star forming region, red represents Seyfert, green is the composite of AGN and star formation and yellow represents Low-Ionization Emission-line Region (LIER).

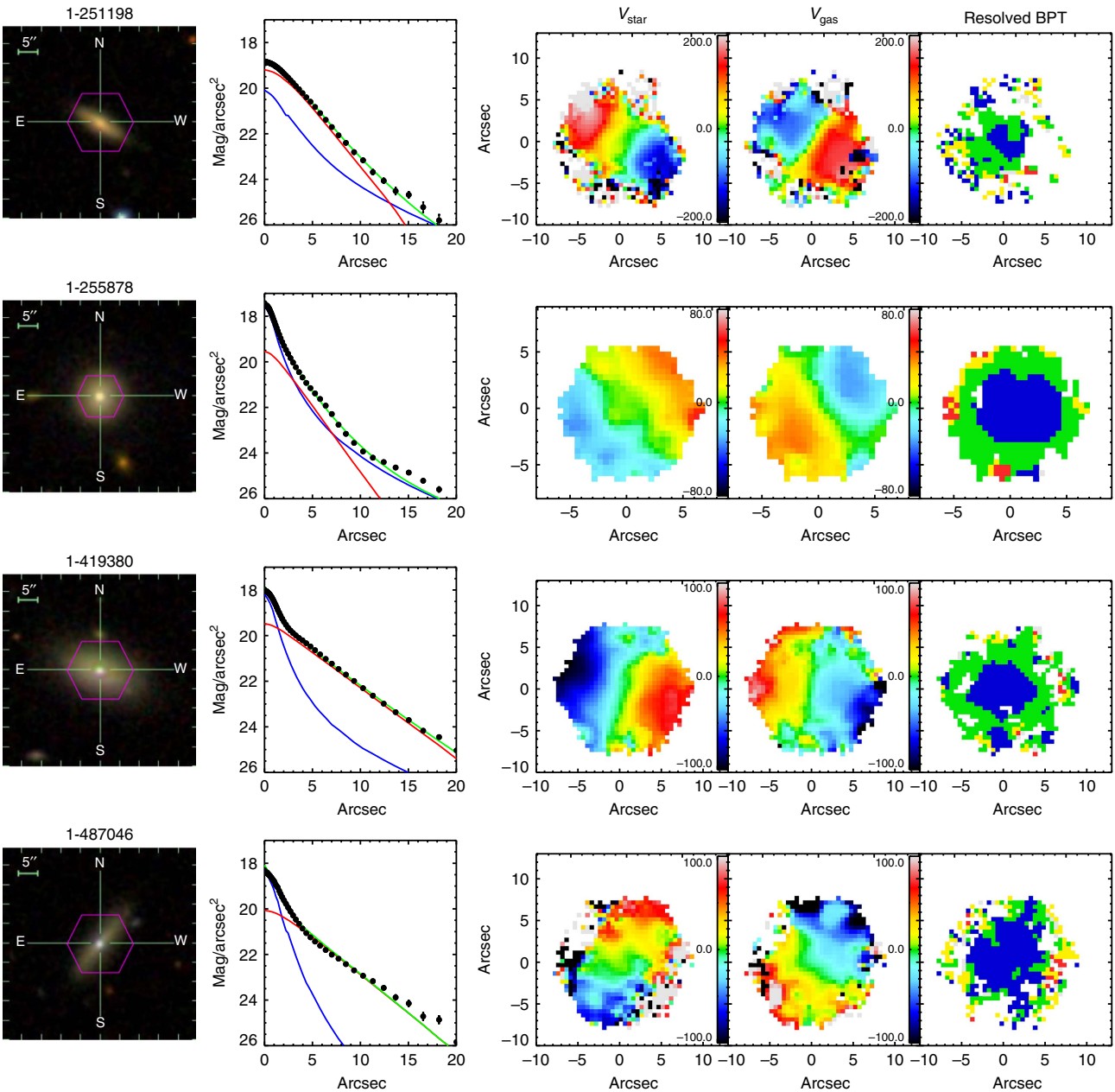

**Figure 7 | Properties of the blue counter rotators.** Same as Fig. 6, but with more objects.

red (red histogram) and green (green histogram) subsamples, as well as the nine blue counter rotators (cyan histogram).

**Sample completeness correction.** An issue with every data set is the selection of weights to correct for missing galaxies. The MaNGA target sample is selected to lie within a redshift range, $z_{min} < z < z_{max}$, that depends on absolute $i$—band magnitude in the case of the Primary and Secondary samples, and absolute $i$—band magnitude and $NUV$—$r$ colour in the case of the colour-enhanced (CE) sample. $z_{min}$ and $z_{max}$ are chosen to yield both the same number density of galaxies and angular size distributions, matched to the IFU sizes, at all absolute $i$—band magnitudes (or magnitudes and colours for the CE sample). This results in lower, and narrower, redshift ranges for less luminous galaxies and higher and wider redshift ranges for more luminous galaxies.

At a given $M_i$ (or $M_i$ and $NUV$—$r$ colour for the CE sample) the sample is effectively volume limited in that all galaxies within $z_{min}(M_i) < z < z_{max}(M_i)$ are targeted irrespective of their other properties. However, that volume varies with $M_i$. Therefore in any analysis of the properties of MaNGA galaxies as a function of anything other than $M_i$ we must correct for this varying selection volume, $V_s(M_i)$—the volume with $z_{min}(M_i) < z < z_{max}(M_i)$. The simplest approach is just to correct the galaxies back to a volume-limited sample by applying a weight ($W$) to each galaxy in any calculation such that $W = V_r/V_s$, where $V_r$ is an arbitrary reference

volume. Since the $z_{min}$ and $z_{max}$ for each MaNGA galaxy are provided in the MaNGA sample catalogue (Wake *et al.*, in preparation), we can easily estimate the fraction of galaxies with decoupled gas and star kinematics in a complete sample by applying this volume correction.

**Global SFR and $M_*$.** Combining SDSS and wide-field infrared survey explorer photometry for the full SDSS spectroscopic galaxy sample, the spectral energy distributions that cover $\lambda = 0.4$–$22\,\mu m$ has been created for a sample of 858,365 present-epoch galaxies[18]. Using MAGPHYS[34], they then model both the attenuated stellar spectral energy distributions and the dust emission at 12 and $22\,\mu m$, producing new calibrations for monochromatic mid-IR SFR proxies, as well as $M_*$.

**Spatially resolved SFR and $M_*$.** Principal component analysis (PCA) is a standard multivariate analysis technique, designed to identify correlations in large data sets. Using PCA, a new method[35] has been generated to estimate stellar masses, mean stellar ages, star formation histories, dust extinctions and stellar velocity dispersions for galaxies from Baryon oscillation spectroscopic survey . To obtain these results, we use the stellar population synthesis models of BC03 (ref. 36) to generate a library of model spectra with a broad range of star formation histories, metallicities, dust extinctions and stellar velocity dispersions. The PCA is run on this library to identify its principal components (PC) over a certain rest-frame

wavelength range $3,700 - 5,500$ Å. We then project both the model spectra and the observed spectra onto the first seven PCs to get the coefficients of the PCs, which represents the strength of each PC presented in the model or observed spectra. We derive statistical estimates of various physical parameters by comparing the projection coefficients of the observed galaxy to those of the models as follows. The $\chi^2$ goodness of fit of each model determines the weight $\sim \exp(-\chi^2/2)$ to be assigned to the physical parameters of that model, when building the probability distributions of the parameters of the given galaxy. The probability density function of a given physical parameter is thus obtained from the distribution of the weights of all models in the library. We characterize the probability density function using the median and the 16–84% range (equivalent to $\pm 1\sigma$ range for Gaussian distributions). In this work, we directly apply this PCA method to the MaNGA data to get the stellar mass for each spaxel.

The SFR for each spaxel is derived from the dereddened Hα luminosity ($L_{H\alpha}$) as SFR $(M_{\odot}\ \mathrm{yr}^{-1}) = 7.9 \times 10^{-42}\ L_{H\alpha}(\mathrm{erg\,s}^{-1})$ to be used Balmer decreasement for dust extinction correction.

**Environment.** We characterize the environment with two parameters, the neighbour number ($N$) and the tidal strength parameter $Q_{lss}$. The neighbour number is defined as the count of galaxies brighter than $-19.5$ mag in $r$-band absolute magnitude within a fixed volume of 1 Mpc in projected radius and $500\ \mathrm{km\,s}^{-1}$ in redshift to the primary galaxy. Given the neighbour number is independent of the stellar mass and cannot account for the interaction a galaxy suffering from its satellites, we also use the tidal strength parameter $Q_{lss}$ to depict the effect of total interaction strength produced by all the neighbours within the fixed volume[22,37]; the higher the parameter, the stronger the interaction. The parameter $Q_{lss}$ is defined as

$$Q_{lss} \equiv \log\left[\sum_i \frac{M_i}{M_p}\left(\frac{D_p}{d_i}\right)^3\right] \tag{1}$$

where $M_i$ and $M_p$ are the stellar masses of the $i^{th}$ neighbour and the primary galaxy. $d_i$ is the projected distance from the primary galaxy to the $i^{th}$ satellite and $D_p$ is the estimated diameter of the central galaxy[22]. Both the number of neighbours and $Q_{lss}$ are drawn from the catalogue generated by Argudo-Fernández et al.

**Surface brightness profile.** We fit the surface brightness profiles of the nine blue counter rotators with three different models: (1) single Sersic; (2) double Sersic; (3) Sersic bulge + exponential disk. The best fitting results are shown in Figs 6 and 7.

**Data availability.** The data supporting the findings of this study are available through SDSS Data Release Thirteen which can be downloaded from http://www.sdss.org/dr13/manga/.

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

## Acknowledgements

Y.M.C. acknowledges support from NSFC grant 11573013, 11133001, the Natural Science Foundation of Jiangsu Province grant BK20131263, the Opening Project of Key Laboratory of Computational Astrophysics, National Astronomical Observatories, Chinese Academy of Sciences. Y.S. acknowledges support from NSFC grant 11373021, the CAS Pilot-b grant no. XDB09000000 and Jiangsu Scientific Committee grant BK20150014. C.A.T. acknowledges support from National Science Foundation of the United States grant no. 1412287. Funding for the Sloan Digital Sky Survey IV has been provided by the Alfred P. Sloan Foundation, the U.S. Department of Energy Office of Science and the Participating Institutions. SDSS-IV acknowledges support and resources from the Center for High-Performance Computing at the University of Utah. The SDSS web site is www.sdss.org. SDSS-IV is managed by the Astrophysical Research Consortium for the Participating Institutions of the SDSS Collaboration including the Brazilian Participation Group, the Carnegie Institution for Science, Carnegie Mellon University, the Chilean Participation Group, the French Participation Group, Harvard-Smithsonian Center for Astrophysics, Instituto de Astrofísica de Canarias, The Johns Hopkins University, Kavli Institute for the Physics and Mathematics of the Universe (IPMU)/University of Tokyo, Lawrence Berkeley National Laboratory, Leibniz Institut für Astrophysik Potsdam (AIP), Max-Planck-Institut für Astronomie (MPIA Heidelberg), Max-Planck-Institut für Astrophysik (MPA Garching), Max-Planck-Institut für Extraterrestrische Physik (MPE), National Astronomical Observatory of China, New Mexico State University, New York University,

University of Notre Dame, Observatório Nacional/MCTI, The Ohio State University, Pennsylvania State University, Shanghai Astronomical Observatory, United Kingdom Participation Group, Universidad Nacional Autónoma de México, University of Arizona, University of Colorado Boulder, University of Oxford, University of Portsmouth, University of Utah, University of Virginia, University of Washington, University of Wisconsin, Vanderbilt University and Yale University.

## Author contributions

Y.M.C. discovered these sources, studied their properties and led the writing of the manuscript. Y.S., C.A.T., M.B., M.M. and E.E. provided the picture to explain all the observation results. Y.S. and M.M. also helped the writing of the manuscript. Y.F.J. helped in making plots. S.H. fitted the surface brightness profile. D.A.W. helped with the sample completeness correction. D.S. searched for deeper images of these galaxies. L.L and M.A.F. provided the environment parameters. K.B., R.B.Y., M.B., N.D., D.A.W., D.T. and D.B. contributed to the design and execution of the survey. All authors commented on the manuscript and contributed to the interpretation of the observations.

## Additional information

**Competing financial interests:** The authors declare no competing financial interests.

**Publisher's note**: 

