## [Peer Review File · Nature Communications]

Reviewers' comments:

Reviewer #1 (Remarks to the Author):

The paper by Chen et al. provides evidence for the existence of counterrotation between gas and stars in blue star forming galaxies (namely galaxies along the main sequence). The analysis of the survey data appears quite exhaustive and robust. Counterrotation has normally been studied for early-type galaxies, and most often for stellar vs. stellar components rather than gas vs. stars, hence in this sense the paper provides a new perspective and adds to the existing literature on the subject. However, the main issue I see is that they find only a very small fraction of their star forming blue galaxies that displays evidence of counterrotation -- 9 out of 489 objects is only 2%, which makes the phenomenon rather irrelevant for galaxy evolution. The fact that it may trigger enhance star formation, and that it affects the relation between stellar velocity dispersion and star formation rate density, is all sound and believable, but again one is left wondering what is the relevance in the global picture of galaxy evolution. Therefore I believe this paper might be of interest for a specialized publication but not be of sufficiently high impact to warrant publication on Nature Communications. One interesting aspect that partially emerges is that the subset showing counterrotation is not associated with major mergers or strong interactions. The authors allude to the possibility that what they see is the result of gas accreting from the cosmic web with misaligned angular momentum. This is indeed seen often in numerical simulations of galaxy formation, and still awaits an observational smoking gun as a test. Had the authors provided one or two examples that this is what they are really seeing I would have marked the paper higher as it would have been a qualitative step forward in testing theoretical models of galaxy formation. However with their current analysis they cannot exclude that what they see is the result of minor mergers with misaligned angular momentum, which is also expected to happen and is not new. In summary, while the paper is of good quality and interesting, it would not have that broad impact expected for publications on Nature Communications.

Reviewer #2 (Remarks to the Author):

I have read the paper and here are my comments:

A: The authors present new results about the frequency of gas counterrotation in star forming galaxies. They conclude that the interaction with native gas funnel all the gas toward the nucleus, building the central regions with new stars.

B: These new results may help to understand the formation and the growth of the galaxy structures in early epoch of the Universe.

F: I suggest some minor revisions: I believe that the paper should also be understandable to readers that do not have specific knowledge of the argument, so adding the following details may help:

- page 5, Discussion, second paragraph: a $\Delta\sigma$ of 20 km/s is not big; I suggest that the authors indicate how is the value of the fluctuations in the velocity dispersion profiles in the sample galaxies and how big are the measurement errors.

- page5, Discussion, third paragraph: α parameter should be better defined here, even if it is already explained in a paper listed as a reference.
- In general, what is the z of the sample galaxies and of the CR blue galaxies? An histogram of N vs. z of the sub-samples may help.

Reviewer #3 (Remarks to the Author):

It is a very interesting paper on internal dynamics and motion of gas in central regions of spiral galaxies. Results were obtained with the new survey MaNGA using integral-field units. This is the state of the art in the field. The paper focuses on few examples of relatively rare, but potentially insightful cases where gas and stars counterrotate. Counter-rotating gas is quite well known in early-type galaxies, but it is rare in late-type spirals. This is the first time the effect is systematically investigated in a rather large sample of galaxies. Results are clearly important for understanding how galaxies form and evolve. The paper will be interesting for a wide range of astronomers (galaxy formation, cosmology) and for general scientific public.

I have few comments on the main results of the paper. It is clearly written and presented in a concise way. My main concerns are about theoretical 'Discussions' section, which I find of poor quality and not acceptable for publication. However, that can be easily improved by the authors by removing the offending simplistic model that authors do not really need in this largely observational paper.

There is some confusion regarding what is considered internal and what is external processes. The authors are coming from community where 'galaxy' means central light-producing region and 'external' is everything outside. This is different for larger group of extragalactic astronomers, people working on the large-scale structure and galaxy formation. For them 'internal' means everything inside the virial radius. In order to avoid the confusion I recommend the authors make this distinction from the start. When in the abstract the authors state that 'the result suggests that the acquisition of abundant external gas ... is possible', is it possible that the gas may come from a satellite galaxy or from gas in the halo? That would be internal gas accretion. How do they know that the gas is really from a cosmological cold filament or from a recently accreted dwarf ("external") galaxy?

Abstract: "This is an important mechanism that may build the central regions of galaxies at high- z where external gas acquisition is much more frequent". This just adds to the confusion. If the counter-rotating gas in $z=0$ spirals is NOT from cold filaments (we really do not know), then this 'mechanism' has nothing to do with the expectation that cold filaments are the main venue for gas accretion at high redshifts. Would be great if the authors clarify the situation.

My main concerns are with the simplistic model that authors put forward in the Discussions section.

"ii) Cancellation of angular momentum occurs from gas–gas collisions between the pre-existing and the accreted gas largely accelerates gas inflow, leading to a fast centrally-concentrated star formation." And later: " ... cancellation of angular momentum and dissipation of kinetic energy through dynamical friction, giving rise to increased velocity dispersion of the ionized gas."

Collision of two gas streams with the relative velocity of $\sim 100\text{km/s}$ produces a shock wave that generates heat with temperature of about a million K. It does not increase velocity dispersion of ionized gas. I understand that the increased gas velocity is observed, but gas physics is a stubborn thing: cannot break it even when it is desirable. Gas heated by shocks may cool very fast. Difficult to estimate it from numbers provided in the paper, but it is likely. So, if gas cools and forms stars then active SF may drive σ_{gas} , but, as the authors argue, the observed rates of SF are too low to drive the large observed σ_{gas} . So, we do not have an explanation of what is actually going on there. That is in striking contradiction of authors statement that 'this scenario to simply explain the above observational facts'.

In any case, dynamical friction cannot produce 'dissipation of kinetic energy'. I do not see how gas can get into tight massive clouds that would suffer from dynamical friction. If the authors have something in mind they should give us their estimates. I suggest they remove this statement.

"Higher metallicity is due to the enrichment from star formation. In a closed-box model, the metallicity will mainly depend on the gas mass fraction f_{gas} ($\equiv M_{\text{gas}}/(M_{\text{gas}} + M_{\text{stars}})$), so the abundances get elevated instantaneously as a large fraction of the available gas turns into stars".

But at the same time, 'external' gas likely had low metallicity. Just turning some fraction of that gas into stars does not increase its metallicity (instantaneously or not). Remember, this is not a closed box by construction. Could authors make their argument slightly more realistic? Simply stating that 'it works' is not enough.

"The infall timescale is given by $t_{\text{infall}} = R_{\text{bulge}}/V_{\text{infall}} = 2.35 \times 10^8 \alpha^{-1} R^{3/2} M^{-1/2}$ yr, where $\alpha = a/0.2$ from numerical simulations (refs 25, 28)."

No, the authors cannot use these 'simulations'. The first is a 20-years old SPH run that does not have large viscosity (but has all

problems of first-generation sph). The second is axysymmetric simulation of accretion disks with alpha-viscosity. This has nothing to do with our problem. Do authors really think that galactic gaseous disks can be treated with alpha disks as in the accretion on the black holes? I do not think so.

Where are indications that "accretion of counter-rotating gas into disk galaxies is an effective way to grow the central bulge"? Of 9 galaxies in the sample, 4 do not have a bulge (1-337898, 1-137890, 1-134004, 1-251198). This is about 1/2 of the sample. From what I see, the counter-rotating spirals are neutral regarding bulges. My guess is that if I take a random sample of spiral galaxies, I will find a larger fraction of galaxies with bulges with the same stellar masses as in the authors sample. I suggest the authors re-write this statement and give more accurate account of the frequency of bulges in counter-rotating spirals.

Reply to Referees' Report on

“The growth of the central region by acquisition of counter-rotating gas in star-forming galaxies”
ref. NCOMMS-16-05862

We are very grateful to all the referees for the very helpful report. We revised and improved the manuscript by including *all* the suggestions and comments from this report. Detailed explanations are given just below each comment or suggestion.

[Comments from the first referee]

The paper by Chen et al. provides evidence for the existence of counterrotation between gas and stars in blue star forming galaxies (namely galaxies along the main sequence). The analysis of the survey data appears quite exhaustive and robust. Counterrotation has normally been studied for early-type galaxies, and most often for stellar vs. stellar components rather than gas vs. stars, hence in this sense the paper provides a new perspective and adds to the existing literature on the subject. However, the main issue I see is that they find only a very small fraction of their star forming blue galaxies that displays evidence of counterrotation – 9 out of 489 objects is only 2%, which makes the phenomenon rather irrelevant for galaxy evolution. The fact that it may trigger enhance star formation, and that it affects the relation between stellar velocity dispersion and star formation rate density, is all sound and believable, but again one is left wondering what is the relevance in the global picture of galaxy evolution. Therefore I believe this paper might be of interest for a specialized publication but not be of sufficiently high impact to warrant publication on Nature Communications. One interesting aspect that partially emerges is that the subset showing counterrotation is not associated with major mergers or strong interactions. The authors allude to the possibility that what they see is the result of gas accreting from the cosmic web with misaligned angular momentum. This is indeed seen often in numerical simulations of galaxy formation, and still awaits an observational smoking gun as a test. Had the authors provided one or two examples that this is what they are relays seeing I would have marked the paper higher as it would have been a qualitative step forward in testing theoretical models of galaxy formation. However with their current analysis they cannot exclude that what they see is the result of minor mergers with misaligned angular momentum, which is also expected to happen and is not new. In summary, while the paper is of good quality and interesting, it would not have that broad impact expected for publications on Nature Communications.

We agree with the referee that 2% is a small fraction, however, we would like to suggest this phenomenon might be important in galaxy evolution for the following reasons:

1) 2% is a lower limit since (1) for the face on galaxies, we can not figure out whether the gas and stars are counter rotating or not from the projected velocity fields. This fraction increases to $\sim 2.5\%$ if we neglect the face on galaxies from our sample; (2) if we assume the mass (or angular momentum) of the accreted gas has a flat distribution, we would expect to find some galaxies in our sample which show positive D4000 gradients but not counter rotators. In these galaxies, the physical process discussed in the paper (redistribution of angular momentum occurs from gas–gas collisions between the pre-existing and the accreted gas largely accelerates gas inflow, leading to a fast centrally-concentrated star formation) also happens, however, we can not detect counter rotating phenomenon since the angular momentum is dominated by the pre-existing gas.

2) we checked all the 489 MaNGA galaxies located along the star forming main sequence, finding that about 5% of them have positive D4000 gradients (indicating younger stellar populations at the central regions and possibly fast growth of the central regions). Thus, the star forming gas-star counter rotators contribute $\sim 40\%$ to the galaxies which are growing the central parts in the local universe. Again this $\sim 40\%$ is a lower limit as we explained in 1).

3) the external gas acquisition is known to be a widely phenomena at high- z , as supported by both observations and theoretical works. The ionized gas kinematics of star forming galaxies at redshift $2\sim 3$

(Genzel et al. 2006, 2008, 2014; Stark et al. 2008; Wright et al. 2009; Law et al. 2009; Puech 2010; Jones et al. 2010; Epinat et al. 2012), as well as their rest-frame optical/UV brightness distributions (Wuyts et al. 2011a), suggest that 30%~70% of the massive ($\log M_*/M_\odot \geq 9.5$) main-sequence star forming galaxies are rotationally supported disks, albeit with frequent perturbations due to minor mergers. The most recent generation of cosmological galaxy evolution models and simulations find that the buildup of $z \geq 1$ star forming galaxies is dominated by smooth accretion of gas and/or minor mergers (Dekel & Birnboim 2006, Guo & White 2008). Our findings that the counter-rotators account for nearly 40% of local galaxies with ongoing growth of the central regions, may thus imply that counter-rotating gas acquisition could be an important mechanism for high- z galaxies to grow the central regions, e.g. blue/red nuggets or high- z compact galaxies. This idea is supported by recent cosmological simulations as well as observational results. Based on cosmological simulations, Zolotov et al. (2015) suggests that the compaction of high- z galaxies is triggered by an intense inflow episode, involving (mostly minor) mergers, counter-rotating streams or recycled gas. Based on sub-kpc ALMA imaging of six star forming galaxies at $z \sim 2.5$, Barro et al. (2005) finds that these high- z galaxies are experiencing nuclear starbursts, the effective radius of the mean SFR profile ($r_{e,\text{SFR}} \sim 1$ kpc) is $\sim 30\%$ smaller than that of the stellar mass profile. Since it's technically impossible to find such blue counter rotators in high- z galaxies, these counter rotators at low- z provide unique insights.

[Comments from the second referee]

I have read the paper and here are my comments:

A: The authors present new results about the frequency of gas counterrotation in star forming galaxies. They conclude that the interaction with native gas funnel all the gas toward the nucleus, building the central regions with new stars.

B: These new results may help to understand the formation and the growth of the galaxy structures in early epoch of the Universe.

F: I suggest some minor revisions: I believe that the paper should also be understandable to readers that do not have specific knowledge of the argument, so adding the following details may help:

(1) page 5, Discussion, second paragraph: a $\Delta\sigma$ of 20 km s^{-1} is not big; I suggest that the authors indicate how is the value of the fluctuations in the velocity dispersion profiles in the sample galaxies and how big are the measurement errors.

Done, we have added these information into Discussions.

(2) page5, Discussion, third paragraph: α parameter should be better defined here, even if it is already explained in a paper listed as a reference.

We have deleted this part according to the comment of the third referee.

(3) In general, what is the z of the sample galaxies and of the CR blue galaxies? An histogram of N vs. z of the sub-samples may help.

Done, we have added the plot to Methods.

[Comments from the third referee]

It is a very interesting paper on internal dynamics and motion of gas in central regions of spiral galaxies. Results were obtained with the new survey MaNGA using integral-field units. This is the state of the art in the field. The paper focuses on few examples of relatively rare, but potentially insightful cases where gas and stars counterrotate. Counter-rotating gas is quite well known in early-type galaxies, but it is rare in late-type spirals. This is the first time the effect is systematically investigated in a rather large sample of galaxies. Results are clearly important for understanding how galaxies form and evolve. The paper will be interesting for a wide range of astronomers (galaxy formation, cosmology) and for general scientific public.

I have few comments on the main results of the paper. It is clearly written and presented in a concise way. My main concerns are about theoretical 'Discussions' section, which I find of poor quality and not acceptable for publication. However, that can be easily improved by the authors by removing the offending simplistic model that authors do not really need in this largely observational paper.

(1) There is some confusion regarding what is considered internal and what is external processes. The authors are coming from community where 'galaxy' means central light-producing region and 'external' is everything outside. This is different for larger group of extragalactic astronomers, people working on the large-scale structure and galaxy formation. For them 'internal' means everything inside the virial radius. In order to avoid the confusion I recommend the authors make this distinction from the start. When in the abstract the authors state that 'the result suggests that the acquisition of abundant external gas ... is possible', is it possible that the gas may come from a satellite galaxy or from gas in the halo? That would be internal gas accretion. How do they know that the gas is really from a cosmological cold filament or from a recently accreted dwarf ("external") galaxy?

We explain what external and internal processes are in the third sentence of the first paragraph.

(2) Abstract: "This is an important mechanism that may build the central regions of galaxies at high- z where external gas acquisition is much more frequent". This just adds to the confusion. If the counter-rotating gas in $z=0$ spirals is NOT from cold filaments (we really do not know), than this 'mechanism' has nothing to do with the expectation that cold filaments are the main venue for gas accretion at high redshifts. Would be great if the authors clarify the situation.

Yes, it is true we can not figure out the origin of the counter rotating gas (from gas rich dwarf galaxies or from cosmic web) from the current data. When we say "external gas acquisition", we mean two possibilities (1) gas acquisition from a gas-rich dwarf galaxy; and (2) gas accretion from cosmic web.

My main concerns are with the simplistic model that authors put forward in the Discussions section.

(3) "ii) Cancellation of angular momentum occurs from gas-gas collisions between the pre-existing and the accreted gas largely accelerates gas inflow, leading to a fast centrally-concentrated star formation." And later: "... cancellation of angular momentum and dissipation of kinetic energy through dynamical friction, giving rise to increased velocity dispersion of the ionized gas."

Collision of two gas streams with the relative velocity of 100km/s produces a shock wave that generates heat with temperature of about a million K. It does not increase velocity dispersion of ionized gas. I understand that the increased gas velocity is observed, but gas physics is a stubborn thing: cannot break it even when it is desirable. Gas heated by shocks may cool very fast. Difficult to estimate it from numbers provided in the paper, but it is likely. So, if gas cools and forms stars then active SF may drive σ_{gas} , but, as the authors argue, the observed rates of SF are too low to drive the large observed σ_{gas} . So, we do not have an explanation of what is actually going on there. That is in striking contradiction of authors statement that 'this scenario to simply explain the above observational facts'.

Collisions between clumpy turbulent media will not just lead to the hot gas that the referee argues, but will leave a (decelerated) cold component with a somewhat larger turbulent velocity dispersion as we argue. Some support for this (in a slightly different context) is in Figure 6 of Matsumoto et al. (2015, ApJ, 801, 77) – compare LT10F0 in left panel to LT10F30 in right panel.

(4) In any case, dynamical friction cannot produce 'dissipation of kinetic energy'. I do not see how gas can get into tight massive clouds that would suffer from dynamical friction. If the authors have something in mind they should give us their estimates. I suggest they remove this statement.

Thanks very much for pointing this out. It is true the mention of dynamical friction is mistaken as we are not claiming large amounts of self-gravity. We have deleted this statement from Discussions.

(5) "Higher metallicity is due to the enrichment from star formation. In a closed-box model, the

metallicity will mainly depend on the gas mass fraction $f_{\text{gas}} (\equiv M_{\text{gas}}/(M_{\text{gas}} + M_{\text{stars}}))$, so the abundances get elevated instantaneously as a large fraction of the available gas turns into stars”.

But at the same time, ‘external’ gas likely had low metallicity. Just turning some fraction of that gas into stars does not increase its metallicity (instantaneously or not). Remember, this is not a closed box by construction. Could authors make their argument slightly more realistic? Simply stating that ‘it works’ is not enough.

Done, we changed this description, please see the first paragraph of Discussions.

(6) “The infall timescale is given by $t_{\text{infall}} = R_{\text{bulge}}/V_{\text{infall}} = 2.35 \times 10^8 \alpha_{0.2}^{-1} R_{5\text{kpc}}^{3/2} M_{10}^{-1/2}$ yr, where $\alpha_{0.2} = \alpha/0.2$ from numerical simulations (refs 25, 28).”

No, the authors cannot use these ‘simulations’. The first is a 20-years old SPH run that does not have large viscosity (but has all problems of first-generation sph). The second is axysymmetric simulation of accretion disks with alpha-viscosity. This has nothing to do with our problem. Do authors really think that galactic gaseous disks can be treated with alpha disks as in the accretion on the black holes? I do not think so.

Many thanks for pointing this out. We can not simply apply alpha-viscosity from accretion disks to our current situation, we have deleted this part as the referee suggested.

(6) Where are indications that ”accretion of counter-rotating gas into disk galaxies is an effective way to grow the central bulge”? Of 9 galaxies in the sample, 4 do not have a bulge (1-337898, 1-137890, 1-134004, 1-251198). This is about 1/2 of the sample. From what I see, the counter-rotating spirals are neutral regarding bulges. My guess is that if I take a random sample of spiral galaxies, I will find a larger fraction of galaxies with bulges with the same stellar masses as in the authors sample. I suggest the authors re-write this statement and give more accurate account of the frequency of bulges in counter-rotating spirals.

From Figure 4, for all the 9 galaxies, the central parts grow faster than the outskirts by about an order of magnitude. It is true we can not call the central parts of these galaxies as bulge since they are disks in the sense of kinematics. We change the sentence into “accretion of counter-rotating gas into disk galaxies is an effective way to grow the central region”.

REVIEWERS' COMMENTS:

Reviewer #1 (Remarks to the Author):

The authors have addressed only partially my comments. In particular they have missed to reply to one of the major remarks that I had, which is crucial for the suitability of the paper to be published on Nature Communications. Indeed I was expecting them to clarify what is the evidence that they are really detecting counterrotating gas streams from the cosmic web as opposed to gas accreted through minor mergers that are counterrotating relative to the disk. Indeed for the latter there is already various evidence at low redshift (albeit not in a large sample analyzed in a systematic way like here), while there is none for the latter as far as I know despite that this is expected from theory/simulations. In the first case the paper would be novel enough to warrant publication in a multi-disciplinary high impact journal, in the second I think this is very good incremental work which would be readily published on MNRAS or ApJ.

Let me also resonate with the remarks of the third referee on the poor theoretical discussion, in particular on the misleading way they talk about the possible role of gas streams together with alpha-disk models. I also think they do not recognize how different are the conditions of galaxy formation between high and low redshift from the way they reply to my report - I would definitely just avoid talking about accretion in high- z galaxies and observation work on it since we they may be comparing apples with oranges. Yet to be fair they made an effort to address the theoretical shortcomings so this is not such a crucial issue anymore.

I would be happy to review the paper again if they can address my concern above but I cannot recommend publication of the paper in the present form.

Reviewer #3 (Remarks to the Author):

The authors addressed my concerns in a satisfactory way. I do not have more comments. The paper is ready for publication in Nature.

Reply to Referees' Report on

“The growth of the central region by acquisition of counter-rotating gas in star-forming galaxies”
ref. NCOMMS-16-05862A

Thanks for the referees in helping us in improving the manuscript. Detailed explanations are given just below each comment or suggestion.

[Comments from the first referee]

(1)The authors have addressed only partially my comments. In particular they have missed to reply to one of the major remarks that I had, which is crucial for the suitability of the paper to be published on Nature Communications. Indeed I was expecting them to clarify what is the evidence that they are really detecting counter-rotating gas streams from the cosmic web as opposed to gas accreted through minor mergers that are counter-rotating relative to the disk. Indeed for the latter there is already various evidence at low redshift (albeit not in a large sample analyzed in a systematic way like here), while there is none for the latter as far as I know despite that this is expected from theory/simulations. In the first case the paper would be novel enough to warrant publication in a multi-disciplinary high impact journal, in the second I think this is very good incremental work which would be readily published on MNRAS or ApJ.

Answer: With the current data, we can not conclude whether the counter-rotating gas streams come from the cosmic web or minor mergers. The first sentence of Discussion part says: “The progenitor accretes counter-rotating gas from a gas-rich dwarf or cosmic web”. However, figuring out the origin of accreted gas is the next step of our research, we have already observed deep images (reach $r \sim 28$ mag/sq. arcsec) of two blue star forming counter rotators with APO 3.5m telescope and more observation time has been approved. Hopefully, we can have some idea about whether the accreted gas comes from cosmic web or minor merger through looking for tidal features.

(2)Let me also resonate with the remarks of the third referee on the poor theoretical discussion, in particular on the misleading way they talk about the possible role of gas streams together with alpha-disk models. I also think they do not recognize how different are the conditions of galaxy formation between high and low redshift from the way they reply to my report - I would definitely just avoid talking about accretion in high- z galaxies and observation work on it since we they may be comparing apples with oranges. Yet to be fair they made an effort to address the theoretical shortcomings so this is not such a crucial issue anymore.

Answer: I agree with the referee that the $z \geq 3$ galaxies are very different from what we find in the manuscript. Thus we changed “high- z galaxies” in the manuscript into “galaxies at $z \sim 1 - 2$ where external gas acquisition is much more frequent.” We also want to bring in some of the arguments that the phenomenon of gas-gas collisions is very complicated, however, by identifying and studying these galaxies in this manuscript we can begin to gain a better understanding of the physics.